# Axial Length Shortening and Choroid Thickening in Myopic Adults Treated with Repeated Low-Level Red Light

**DOI:** 10.3390/jcm11247498

**Published:** 2022-12-17

**Authors:** Guihua Liu, Bingqin Li, Hua Rong, Bei Du, Biying Wang, Jiamei Hu, Bin Zhang, Ruihua Wei

**Affiliations:** 1Tianjin Key Laboratory of Retinal Functions and Diseases, Tianjin Branch of National Clinical Research Center for Ocular Disease, Eye Institute and School of Optometry, Tianjin Medical University Eye Hospital, Tianjin 300384, China; 2College of Optometry, Nova Southeastern University, Davie, FL 33314, USA

**Keywords:** repeated low-level red light therapy (RLRL), myopia, adults, axial length, choroidal thickness, choroidal vascularity index

## Abstract

This study aimed to explore the effect of repeated low-level red light (RLRL) on axial length (AL), choroid blood flow, and anterior segment components in myopic adults. Ninety-eight myopic adults were randomly divided into the RLRL group (*n* = 52) and the control group (*n* = 46). Subjects in the RLRL group completed a 4-week treatment composed of a 3-min RLRL treatment session twice daily, with an interval of at least 4 h. Visits were scheduled before and on 7, 14, 21, and 28 days after the treatment. AL, subfoveal choroidal thickness (SChT), choroidal vascularity index (CVI), and anterior segment parameters were measured at each visit. A linear mixed-effects model showed that the AL of the subjects in RLRL decreased from 24.63 ± 1.04 mm to 24.57 ± 1.04 mm, and the SChT thickened by 18.34 μm. CVI had a slight but significant increase in the 0–6 zone. However, all the anterior segment parameters did not change after RLRL treatment. Our study showed that the choroid’s thickening is insufficient to explain the axial length shortening. The unchanged anterior segment and improved choroid blood flow suggest that the AL shortening in this study is mainly related to changes in the posterior segment.

## 1. Background

Myopia is a common ocular disorder that has been dramatically elevated worldwide over the past several decades [1,2]. It is a significant factor for visual impairment, and high myopia can increase the risk of pathologic ocular changes [3]. Axial elongation is an essential indicator of myopia progress. Previous studies have shown that eyes with axial length (AL) of 26 mm or greater had a significantly higher risk for visual impairment than eyes with AL shorter than 26 mm in Europeans under 60 [4]. Therefore, intervention methods have been explored to slow down myopia progression and reduce the incidence of high myopia.

Recently, a new intervention—repeated low-level red light (RLRL) therapy—showed a significant myopia control effect in myopic Chinese children [5,6,7,8,9]. It was administered at home for 3 min per session, twice daily with a minimum interval of 4 h, five days per week. Jiang et al.’s study showed that, compared with the non-treatment group, the RLRL group represented a 69.4% reduction in axial elongation after one year of treatment [5]. More interestingly, they found axial length shortening (>0.05 mm change) in 39.8% of the participants at one month and 21.6% at 12 months. The proportion of children showing axial length shortening may have been underestimated because of the physiological growth in the axial length. Therefore, exploring this phenomenon in adults with stable axial length may be more appropriate.

The mechanism underlying the observed axial length shortening remains unknown. Choroid thickening was found in children who underwent RLRL treatment [5,6]. However, it could not explain the axial shortening entirely as the scale of choroidal thickening was much smaller. An alternative hypothesis is that the RLRL treatment might increase choroid blood flow and metabolism of the fundus, thus ameliorating scleral hypoxia and restoring scleral collagen levels [5]. However, the blood flow changes of the fundus in myopic adults after RLRL treatment have not been examined. Changes in other ocular components could also contribute to the short-term axial length change. Hughes reported that anterior chamber depth (ACD), crystalline lens thickness (LT), anterior segment length (ASL), and vitreous chamber depth (VCD) altered significantly during accommodation, resulting in an increment in axial length [10]. Moreover, such changes were more prominent in the myopic group and existed in adults and children [10,11,12]. Whether short-term RLRL treatment can cause changes in anterior segment parameters in myopic adults is currently unknown.

Therefore, this study aimed to explore whether axial length shortening exists in myopic adults who received RLRL after one month of treatment and to quantify changes in choroid flow and ocular components in the anterior segment.

## 2. Materials and Methods

### 2.1. Subjects

This prospective study was conducted at the Tianjin Medical University Eye Hospital (Tianjin, China) between July 2022 and September 2022. All enrolled participants signed informed consent forms. All procedures adhered to the tenets of the Declaration of Helsinki and were approved by the Tianjin Medical University Eye Hospital ethics committee. Participants underwent a complete ophthalmic examination, including visual acuity, intraocular pressure, slit-lamp ophthalmic examination, and fundus examination. The inclusion criteria were age between 18 and 35 years; myopia refractive error power between −6.00 D and −0.50 D, with the rule of astigmatism less than −1.50D, and best corrected monocular optical visual acuity better than 20/20. The exclusion criteria included acute and sub-acute inflammations or infection of the anterior chamber of the eye, history of surgery or contact lens use, severe insufficiency of tears, corneal hypoesthesia, allergic eye diseases, retinal diseases, and any significant systemic illness. Eligible subjects were randomly allocated to either the RLRL group or the control group, according to a randomization list pre-generated by a computer program. All technicians involved in the study did not know the grouping.

### 2.2. Measurement

Lenstar and SS-OCT were measured before the treatment, seven days, 14 days, 21 days, and 28 days after the treatment. All evaluations were performed from 9 to 11 am to avoid diurnal variation [13].

### 2.3. Repeated Low-Level Red Light Therapy (RLRL) 

A low-level red light therapy device (Eyerising; Suzhou Xuanjia Optoelectronics Technology, Suzhou, China) was used in this study. It consists of semiconductor laser diodes, which deliver low-level red light with a wavelength of 650 ± 10 nm at an illuminance level of approximately 1600 lux through the pupil to the fundus. A recent study has shown it to be safe for a 2-year treatment period [8]. In the control group, a neutral density attenuator was placed between the laser emitter and the eyepiece to lower the red light power by three log units. The average red light power was 1.63 mW and 1.92 μW before and after attenuation, measured by the Laser power meter (UT385, UNI-Trend Technology, Dongguan, China).

Subjects wore spectacles to complete treatment twice daily with an interval of at least 4 h, with each treatment lasting 3 min, during weekdays (5 days per week). All subjects were required to continue treatment for one month. To ensure an accurate measure of compliance, the date and time of treatment sessions were automatically captured by the device and transferred to the server through the internet. A reminder message was sent to both the supervisor and the subject upon noticing that the subject had not logged into the system for two days consecutively. The supervisor would confirm with the subject to ensure the number of treatments required by the test (twice a day, more than 4 h intervals, 5 days a week).

### 2.4. Ocular Biometric Parameters

Ocular biometric parameters were measured with a non-contact optical low-coherence reflectometry device (Lenstar LS-900; Haag-Streit AG, Berne, Switzerland) before and after the treatment. Previous studies reported that Lenstar had good repeatability and stability, and had been used to monitor subtle changes in ocular biometric parameters such as axial length after intervention [14,15,16,17,18]. Subjects were advised to fixate on the target during the measurements and blink several times for an intact tear film. Only the intra-session differences of three repeated measurements no greater than 0.02 mm were kept on each measurement occasion. The average was used as a representative value for further analysis. Biometric parameters that included axial length (AL), center cornea thickness (CCT), anterior chamber depth (ACD), lens thickness (LT), flat meridian (K_1_), and steep meridian (K_2_) were collected.

### 2.5. SS-OCT/OCTA

A swept-source OCT (VG200, SVision Imaging, Ltd., Luoyang, China) was performed for SS-OCT and SS-OCTA images with radial and raster patterns. It contains a central tunable laser with 1050 nm wavelength by 200,000 A scan per second [19]. A cross-pattern light was the indicator for the patient to make the fixation, and the radial and raster patterns were built into the software to capture the images. B-scan (55°) images that were 16 mm-wide demonstrated fundus structures to rule out the abnormal structures. A total of 512 × 512 scans of OCTA images obtained in a 6 mm x 6 mm macular cube pattern were captured by Angio Retina mode for retinal vascularity assessments. The choroidal boundary and large/medium choroidal vessels were accurately segmented in three dimensions using the segmentation algorithm based on deep learning. The subfoveal choroidal thickness (SChT) was defined as the vertical distance between Bruch’s membrane and the choroid−sclera interface in the macular zone. The choroidal vascularity index (CVI) was defined as the ratio of choroidal vascular luminal volume to total choroidal volume (Figure 1). The macular zone was interested in that study, and divided into three circular concentric areas on the base of the Early Treatment Diabetic Retinopathy Study (ETDRS) grid [20]. In this study, three chosen rings ranged from 0 to 1 mm, 0 mm to 3 mm, and 0 mm to 6 mm. A well-trained operator performed all scanning operations, and all the pictures reserved for the study’s signal strength were higher than 7.

## 3. Statistics

The means and standard deviations of measured parameters were computed for descriptive purposes. Data were first tested for normality using the sample Shapiro–Wilk test. Independent t-tests and chi-square tests were used to assess the differences between the RLRL group and control group at baseline. Two-way repeated-measures ANOVA was used to analyze the group, time, and group*time interaction. Pearson correlation was used to analyze the relationship between age, baseline AL, SChT changes, and AL changes. All statistical analyses were performed using R software (version 3.2.2). Only data from the right eye of each subject were included in the analysis. A *p*-value < 0.05 value was defined as statistically significant.

## 4. Results

There were 110 subjects enrolled initially, and 98 subjects completed all follow-up examinations, with 52 subjects (28 males and 24 females; mean age 26.08 ± 2.13 years, range: 23–31 years) in the RLRL group and 46 subjects (26 males and 20 females; mean age 22 years, range: 23–31 years) in the control group. No significant differences in age, sex, SChT, CVI, and all ocular biometric parameters were identified between the two groups at baseline (Table 1).

### 4.1. Change in Axial Length

In the RLRL group, AL was shortened steadily within one month of treatment. By the end of the first month of treatment, AL decreased by an average of 0.06 mm (from 24.63 ± 1.04 mm to 24.57 ± 1.04 mm, range: 0–0.13 mm). In the treated group, 69.23% of subjects’ AL decreased by more than 0.05 mm. In the control group, AL had no significant changes during the follow-up. There was a statistically significant group*time interaction by a two-way repeated ANOVA (*p* < 0.001). In other words, the AL in the RLRL group (red line) decreased gradually with the treatment time, while AL in the control group (blue line) did not change. The post-hoc analyze indicated that there were statistically significant differences between the RLRL group at two different time points (all *p* < 0.001) except the comparison between 21 days and 28 days (*p* > 0.05). There was no significant difference in the control group (all *p* > 0.05). (Figure 2A).

### 4.2. Change in SChT

In the RLRL group, the subfoveal choroidal thickness increased significantly after treatment. The thickening was most significant in the second week of treatment and then increased slowly until the average SChT thickened by 18.34 μm in one month of treatment (from 365.73 ± 81.74 μm to 384.12 ± 78.67 μm). In contrast, there was no significant difference in SChT in each follow-up and control group. Two-way repeated ANOVA analysis showed a significant group*time interaction (*p* < 0.001). This showed that SChT in the RLRL group (red line) increased over time, while there was no change in the control group (blue line) (Figure 2B).

### 4.3. Change in CVI

In this study, we divided the subfoveal CVI into 0–1 mm, 0–3 mm, and 0–6 mm zones referring to the early treatment of diabetic retinopathy study (ETDRS) to evaluate the changes in subfoveal choroidal blood flow from the center to the whole. A two-way repeated ANOVA revealed that the statistically significant CVI_0–6_ changes in RLRL group increased from 0.361 ± 0.049 to 0.375 ± 0.054 after one month of RLRL treatment (group*time interaction *p* = 0.001). However, there was no statistical change in CVI_01_ and CVI_03_ and all zones of the control group (all *p* > 0.05) (Table 2).

### 4.4. Change in Other Ocular Biometric Parameters

Table 3 demonstrates the alterations in anterior segment parameters of the two groups at each follow-up. The two groups’ biological parameters of the anterior segment were similar. During the one-month follow-up, CCT, ACD, LT, K1, and K2 did not change significantly (all *p* > 0.05). RLRL therapy did not cause changes in anterior segment parameters (Table 3).

### 4.5. Relationship between the AL Changes and Other Parameters

Age, baseline AL, and SChT alterations were used to analyze the effect on AL changes after one month of treatment. The results indicated that SChT thickening was significantly associated with AL shortening (r = 0.51, *p* < 0.01), whereas age and baseline AL were not associated with AL changes (*p* = 0.92 and *p* = 0.94, respectively).

## 5. Discussion

In this current study, we found that the AL of myopic adults was significantly shortened after one month of RLRL therapy, along with increased choroid thickness and improved choroid circulation. Since no significant changes were found in the anterior segment, we reasoned that the AL shortening observed in this study was mainly associated with changes in the posterior segment.

### 5.1. Adult Vs. Children Findings

Recent studies have reported that RLRL can effectively retard myopia progress in myopic Chinese children by shortening the axial length [5,6,7]. Xiong et al. reported that, after 6-month of RLRL treatment, the mean axial length change was −0.06 ± 0.15 mm, compared to 0.20 ± 0.06 mm in subjects wearing single-vision spectacles [7]. Zhou et al. reported that at nine months, axial length changes were −0.06 ± 0.19 mm and 0.26 ± 0.15 mm in the treatment group and control group (*p* < 0.001), respectively [6]. Jiang reported that 39.8% of myopic children in the RLRL group at one-month follow-up achieved an axial length shortening of ≥0.05 mm [5]. In this current study, 69.23% of the adult subjects demonstrated axial length shortening ≥ 0.05 mm after one month of RLRL treatment. In children, usually up to 15 years of age, the eyeball is still growing, and the axial length constantly changes [21,22,23]. Therefore, RLRL’s axial length shortening effect may be partially masked by the axial growth and was underestimated by the previous studies.

Among myopic children who received RLRL treatment, elder children with greater baseline axial length tended to have more significant axial length shortening [5,6]. However, in this study’s adult subjects, there was no significant association between axial length change and age or baseline axial length after one month of treatment. One possibility for such a discrepancy may be the short follow-up duration, one month, in this study. The other possibility may be the age-dependent axial length change observed in children. Therefore, a study with a more extended follow-up period is needed in the future.

### 5.2. Choroid Thickened, but Not Enough

Based on existing studies, the observed axial length shortening could be partially attributed to the thickening of the choroid layer. In Xiong’s study, the axial length decreased to 25.01 ± 1.14 mm from 25.06 ± 1.14 mm, while choroid thickness increased to 311.84 ± 67.08 μm from 288.61 ± 59.59 μm [7]. In Jiang’s study, axial length shortened by 0.04 mm on average, while choroid thickness increased by 16.1 μm [5]. This number was very close to the value reported in the present study, an 18.34 μm increase in choroid thickness. Therefore, the effect of RLRL on choroid thickness after one month is very similar in children and adults.

On the other side, the change in choroid thickness cannot fully explain the axial length shortening observed. The secondary changes following increased choroid thickness include increased circulation volume and speeded-up metabolism, improved sclera hypoxia, and recovery of the depleted collagen level [5,6,24,25]. We measured the CVI before and after the RLRL treatment to test if such a hypothesis is true. Our data revealed a slight but significant increase in CVI following one month of treatment of RLRL. This finding is consistent with the idea that improved choroid circulation is the underlying reason for shortened axial length. Moreover, agreeing with other studies [5,6,7], this current study did not find any significant changes in the anterior segment. Therefore, the axial length shortening occurs mainly in the posterior segment of the eye.

### 5.3. Alternative to Natural Outdoor Lights

Outdoor activities and exposure to natural light were closely associated with myopia prevention, which suggests that light exposure plays a protective role in myopia development [26,27]. Therefore, another approach for myopia control could be an alternative to sunlight that can slow down myopia progress. Much debate still exists on the effect of light with different wavelengths on ocular parameters and its relevance to myopia control in animal models and clinical studies [18,28,29,30,31,32]. Red light-induced refractive development alterations were associated with reduced vitreous chamber elongation and increased choroidal thickness in a rhesus monkey’s model [29,32]. However, the results from other species, such as mice and guinea pigs, indicated that violet (380 nm) and blue light (470 nm) inhibited eye growth [31,33]. Thakur et al. reported that, in human subjects, exposure to red lights and green lights led to elongated axial length, while exposure to blue lights led to shortened axial length [28]. Similar conclusions were also confirmed in Lou’s study [30].

Other studies investigated the effect of luminance levels on ocular biometrics. Chakraborty measured the axial length and choroid thickness changes when the human eye was exposed to low luminance (500 Lux), moderate luminance (1000 Lux), and darkroom level (<5 Lux) briefly. Axial length shortened in the low and moderate luminance level, but not at the dark room level. The axial length shortening is much greater with the moderate luminance level (−0.013 mm) when compared to the shortening (−0.006 mm) observed in the low luminance level, although not statistically significant. In future studies, a higher luminance level might be needed to quantify the relationship between luminance level and axial length growth [17]. In animal models, elevated lighting levels significantly reduced the degree of axial myopia induced by form deprivation in chickens, tree shrews, and macaques [34,35,36,37]. In RLRL treatment, more light energy and intensity are projected to the retina when red light enters the eyeball as a laser beam, compared to scattered lights in a lab environment. Future studies should map axial length changes to light within various bandwidths at different luminance levels.

### 5.4. Axial Length Shortening in Orthokeratology

Axial length shortening is not only found in RLRL. Some myopic patients treated with orthokeratology (OK) lenses have also demonstrated axial length shortening [38,39,40]. Lau et al. reported that during the first week of OK lens treatment, the mean AL shortening was 0.026 mm. However, after the first week, axial length changes were not significantly different from the controls wearing spectacles [39]. Therefore, axial length shortening was considered a phenomenon existing only in the early stage of OK treatment. Swarbrick et al. reported a 0.04 mm AL shortening three months after lens-wearing. However, their data found no significant difference from spectacle-wearing controls six months into treatment [40]. The two potential mechanisms leading to shortened axial length in OK lens treatment are thinned cornea and thickened choroid. In RLRL and OK treatment, choroid plays a vital role in myopia control. Whether choroid thickening associated with RLRL could be used in myopia prevention remains to be studied. Moreover, Wang reported that among children who underwent OK treatment, those who showed axial shortening in the early stage often had a better myopia control effect (smaller axial length growth) than those who did not show an axial shortening in the early stage [41]. In this study, axial shortening was significantly correlated with choroid thickening after one month of treatment. Whether choroid thickening at the early stage of RLRL treatment could be utilized as a predictor for long-term success calls for more attention.

Until now, no serious complications have been reported in RLRL’s studies with periods from 6 months to 2 years [6,7,8]. There was even a 12-month study reporting improved accommodative function after RLRL treatment [9]. However, it is unknown whether long-term RLRL treatment could cause retinal structural damage or functional damage. Therefore, more long-term studies are needed to confirm RLRL treatment’s long-term safety. In view of the current research status of RLRL treatment, clinicians should pay attention to the examination results of macular OCT and mERG and other indicators during follow-up. In addition, clinicians should also think highly of the change of afterimage duration after treatment, so as to ensure the safety of the irradiated eye to the greatest extent.

### 5.5. Limitations

Our present study has several limitations. First, the follow-up duration of this study is relatively short. It is not clear whether the subjects have reached the maximal potential amount of axial shortening or choroid thickness. Second, axial length growth rebound has been reported in children. Chen et al. reported that the AL increased by 0.16 mm after three months of treatment cessation in myopic children [9]. Xiong’s study reported a second-year axial elongation of 0.42 mm in the RLRL-SVS group when the RLRL treatment was stopped. This progression rate was greater than the second-year progression (0.28 mm) in the SVS-SVS group [8]. However, we do not know how the axial length will rebound after the RLRL treatment stops in myopic adults. Third, it is not clear how the short-term axial shortening correlates with long-term myopia control. Fourth, it is not clear if such short-term axial shortening also occurs in emmetropic subjects or if it exists only in myopic adults.

## 6. Conclusions

In myopic adults who received one month of RLRL treatment, axial length was significantly shortened, and choroid flow was increased considerably. However, the choroid’s thickening is insufficient to fully explain the axial length shortening. In combination with an unchanged anterior segment, our study suggests that the AL shortening in this study is mainly related to changes in the posterior segment.

## Figures and Tables

**Figure 1 jcm-11-07498-f001:**
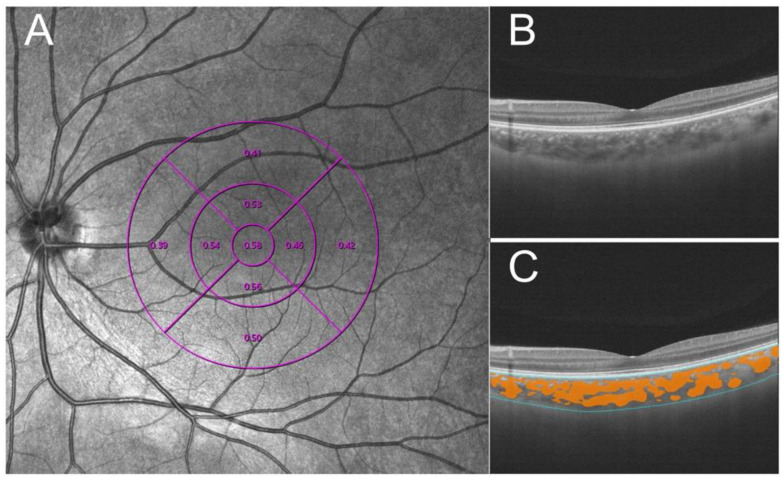
Methods to measure choroidal vascularity index (CVI). (**A**) The ETDRS grid divided the macular region into three concentric rings (1, 3, and 6 mm in diameter [purple lines]). (**B**) Raw image acquired by SS-COT. (**C**) Based on deep learning, the system recognizes the boundary of choroid(blue lines) and identifies the choroid’s large and medium blood vessels (orange zone) and quantifies CVI.

**Figure 2 jcm-11-07498-f002:**
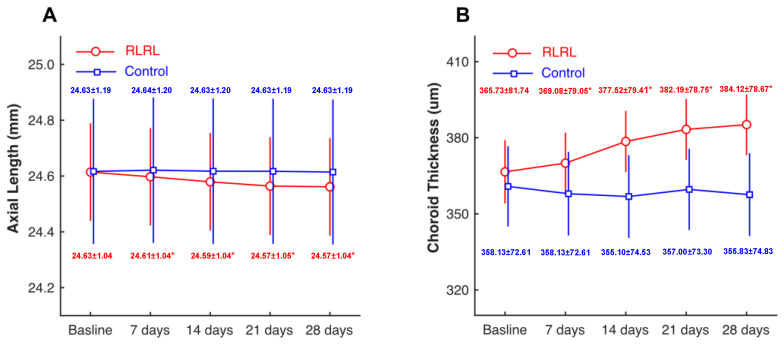
Time course of axial length and choroidal thickness in two groups. Compared with the control group (blue line), the RLRL group (red line) had decreased axial length (**A**) and increased choroid thickness (**B**) after treatment. * *p* < 0.05 when compared to the baseline.

**Table 1 jcm-11-07498-t001:** Demographics and baseline ocular characteristics between the RLRL group and control group (mean ± SD).

Characteristics	RLRL Group (*n* = 52)	Control Group (*n* = 46)	*p*
Age (years)	26.08 ± 2.13	25.60 ± 2.42	0.43
Sex (male/female)	28/24	26/20	0.79
SChT (μm)	365.73 ± 81.74	358.13 ± 72.61	0.63
CVI_01_	0.389 ± 0.054	0.384 ± 0.058	0.66
CVI_03_	0.386 ± 0.050	0.387 ± 0.051	0.11
CVI_06_	0.361 ± 0.049	0.365 ± 0.042	0.26
CCT (μm)	522.77 ± 47.96	519.65 ± 42.83	0.74
ACD (mm)	3.02 ± 0.24	3.05 ± 0.27	0.58
LT (mm)	3.72 ± 0.20	3.69 ± 0.15	0.49
K_1_ (D)	42.85 ± 1.60	42.68 ± 2.08	0.66
K_2_ (D)	44.01 ± 1.82	44.00 ± 2.19	0.97
AL (mm)	24.63 ± 1.04	24.63 ± 1.19	0.98

SChT = subfoveal choroidal thickness; CVI = choroidal vascularity index; CCT = center cornea thickness; ACD = anterior chamber depth; LT = lens thickness; K_1_ = flat meridian; K_2_ = steep meridian; AL = axial length.

**Table 2 jcm-11-07498-t002:** Change in choroidal vascularity index at different periods (mean ± SD).

Characteristics	RLRL Group (*n* = 52)	Control Group (*n* = 46)
CVI_01_		
Baseline	0.389 ± 0.054	0.384 ± 0.058
7 days	0.395 ± 0.052	0.383 ± 0.060
14 days	0.403 ± 0.048	0.384 ± 0.061
21 days	0.402 ± 0.053	0.385 ± 0.060
28 days	0.404 ± 0.053	0.384 ± 0.061
P-Group	0.35	
P-Time	0.16	
P-Group*Time interaction	0.27	
CVI_03_		
Baseline	0.386 ± 0.050	0.387 ± 0.051
7 days	0.390 ± 0.053	0.388 ± 0.050
14 days	0.396 ± 0.053	0.390 ± 0.050
21 days	0.398 ± 0.056	0.390 ± 0.048
28 days	0.398 ± 0.056	0.388 ± 0.048
P-Group	0.62	
P-Time	0.10	
P-Group*Time interaction	0.07	
CVI_06_		
Baseline	0.361 ± 0.049	0.365 ± 0.042
7 days	0.368 ± 0.052	0.365 ± 0.041
14 days	0.369 ± 0.052	0.363 ± 0.042
21 days	0.374 ± 0.055	0.362 ± 0.041
28 days	0.375 ± 0.054	0.363 ± 0.042
P-Group	0.67	
P-Time	0.001	
P-Group*Time interaction	0.001	

CVI = choroidal vascularity index.

**Table 3 jcm-11-07498-t003:** Change in anterior segment parameters at different time points (mean ± SD).

	Characteristics									
	CCT (μm)		ACD (mm)		LT (mm)		K_1_ (D)		K_2_ (D)	
	RLRL	Control	RLRL	Control	RLRL	Control	RLRL	Control	RLRL	Control
Baseline	522.77 ± 47.96	519.65 ± 42.83	3.02 ± 0.24	3.05 ± 0.27	3.72 ± 0.20	3.69 ± 0.15	42.85 ± 1.60	42.68 ± 2.08	44.01 ± 1.82	44.00 ± 2.19
7 days	521.71 ± 47.32	528.87 ± 42.92	3.01 ± 0.27	3.04 ± 0.27	3.72 ± 0.23	3.69 ± 0.17	42.87 ± 1.61	42.64 ± 2.07	44.01 ± 1.85	44.00 ± 2.10
14 days	521.62 ± 47.40	522.11 ± 43.47	3.01 ± 0.25	3.05 ± 0.27	3.72 ± 0.21	3.69 ± 0.16	42.84 ± 1.64	42.58 ± 2.07	43.95 ± 1.85	43.91 ± 2.17
21 days	522.04 ± 47.68	520.89 ± 42.48	3.02 ± 0.25	3.04 ± 0.27	3.72 ± 0.21	3.70 ± 0.17	42.83 ± 1.64	42.59 ± 2.06	43.96 ± 1.88	43.85 ± 2.15
28 days	521.69 ± 48.07	516.57 ± 42.87	3.01 ± 0.24	3.04 ± 0.28	3.71 ± 0.20	3.69 ± 0.16	42.81 ± 1.62	42.66 ± 2.07	44.04 ± 1.80	43.91 ± 2.16
P-Group	0.94		0.97		0.74		0.38		0.64	
P-Time	0.49		0.22		0.72		0.63		0.49	
P-Group * Time	0.19		0.09		0.96		0.69		0.31	

CCT = center cornea thickness; ACD = anterior chamber depth; LT = lens thickness; K_1_ = flat meridian; K_2_ = steep meridian.

## Data Availability

The datasets used and analyzed during the current study are available from the corresponding authors upon reasonable request. The data are not publicly available due to patient privacy.

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
