# Peer review of "Axial Length Shortening and Choroid Thickening in Myopic Adults Treated with Repeated Low-Level Red Light"

_jcm, 2022, doi:10.3390/jcm11247498_

Round 1
Reviewer 1 Report (Previous Reviewer 1)
I consider that the authors have made the suggested changes
Author Response
Thank you
Reviewer 2 Report (Previous Reviewer 2)
The changes provided by the authors were insufficient due the demand by the previous reviewers
The topic of the article is not of the interest of the whole community of ophthalmologist and vision professional
The technology applied do not have the sufficiente background to achieve good results
The sample size is small and the strenght of the manuscript needs to be improve with future and additional research, measures and analysis
Author Response
Using RLRL for myopia control is still in its early days of development and is far from being firmly established. Many related questions still remain to be answered, such as safety and long-term effects. However, exactly due to this developing nature, it is essential to have more data collected, particularly data from different perspectives. Any small piece of data will shed new light on this critical topic contributively. In this sense, our study definitely adds value to the existing literature. We explored a formerly neglected issue of how RLRL affects the axial length and anterior segments in adults instead of myopic children.
Reviewer 3 Report (New Reviewer)
Thank you for this nice paper confirming the efficacy of RLRL to adult myopic patients and investigating the mechanism of reported efficacy in AL shortening by assessing choroidal or anterior segmental structural changes. However, I found some issues that need to be addressed prior publishing. Please find the lists of major comments below.
1.
Authors have to discuss about the subtle changes of AL as treatment efficacy. Although we can find that authors took care about this point by using multiple repeated results of biometry, please provide evidences that show the robustness of AL measurements obtained by Lenstar LS-900 referring the previous literature.
2. Page 7, Line 10-11
Please provide the results of post-hoc analyses to compare the AL measurements on two different time points.
3.
Your RLRL is still emerging treatment, although preceding studies showing promising results have been already published. It is better to disclose the potential adverse effects of this treatment and their managements by introducing the preclinical trials.
In addition, I listed some minor comments below. Please use these for your information.
4. Please confirm the formatting of the Tables.
5. Please confirm the formatting of the references in manuscript body and unify it.
Author Response
>> 1. Authors have to discuss about the subtle changes of AL as treatment efficacy. Although we can find that authors took care about this point by using multiple repeated results of biometry, please provide evidences that show the robustness of AL measurements obtained by Lenstar LS-900 referring the previous literature.
Answer: Related contents and references have been added to page 3, lines 29-30.
>> 2. Page 7, Line 10-11 Please provide the results of post-hoc analyses to compare the AL measurements on two different time points.
Answer: The post-hoc analysis indicated that there were statistically significant between the RLRL group at two different time points (all p< 0.001) except the comparison between 21 days and 28 days (p>0.05). There was no significant difference in the control group (all p>0.05). Related contents have been added to the text (page 5, lines 21-23).
>> 3. Your RLRL is still emerging treatment, although preceding studies showing promising results have been already published. It is better to disclose the potential adverse effects of this treatment and their managements by introducing the preclinical trials.
In addition, I listed some minor comments below. Please use these for your information.
Answer: Thanks for your advice; related content has been added to the text. (page 10, lines 8-16)
>> 4. Please confirm the formatting of the Tables.
Answer: This issue has been confirmed.
>> 5. Please confirm the formatting of the references in the manuscript body and unify it.
Answer: This issue has been confirmed.
Round 2
Reviewer 2 Report (Previous Reviewer 2)
The authors did not solve the comments
The papers did not change enough on behalf the first point
Several methodological issues are still present in the manuscript
Reviewer 3 Report (New Reviewer)
The authors have addressed my points appropriately.
This manuscript is a resubmission of an earlier submission. The following is a list of the peer review reports and author responses from that submission.
Round 1
Reviewer 1 Report
The authors present good work as far as the objectives are concerned.
Before making the analysis of the discussion of the work it should be reviewed in terms of the statistics.
The paper needs to be revised on the following points:
1- Line 47 “Choroid thickening was found in children who underwent RLRL treatment”
Put reference
2- Line 60-62
Put the time of treatment
3- Line 82-83 “All evaluations were performed from 9 to 11 am to avoid diurnal variation.”
Put reference
4- Line 138-144
Why didn't the authors perform repeated measures ANOVA analysis? These should be performed, with the bonferroni correction for example for pairwise analysis.
Results
5- Line 147-148, 98 patients in this phase and 42-30 patients in 3.3 ?
6- line 149-150, “No significant differences in age, sex, SChT, CVI, and all ocular biometric parameters were identified between the two groups at baseline.”
Present a table with these values and the statistical significance of the described test.
7 - Table 2, put the test statistic horizontally on each item (RLRL vs Control) in each period and redo the statistic vertically (RLRL or Control) with repeated-measures ANOVA and pairwise analysis.
Reviewer 2 Report
The authors aimed to explore the effect of repeated RLRL on AL, choroid and anterior segment and they found that choroid get thick, AL get short and anterior segment did not change.
The topic of the research has a limited interest to the potential readers. In this case many ethics issue could involved in this study. The used of this red light, as stated for the authors, is previously use and get safe results. But the device instrument is validated, or the safe aspects was controlled in a short term?
The thickness of the choroid after the RLRL could be due a inflammation of the layer and it have to be studied to confirm it is safe or not. In the same way, the shortened of the axial length is temporal and after the treatment with the RLRL has a rebound effect as the authors has stated on the limitation section.
The research has limited results achieved with the modification of the ocular parameters and the potential side effects of the RLRL was not studied and could imply a danger to the retina or visual function of the patients.
The authors should include ocular safe variables to confirm that this technology could be used to reduce the axial length and in order to reduce the future myopia. At this moment, this technology has not the sufficient scientific literature background to stablish as a technology to used as myopia control.
The results was not presented clearly and additional graphs would improve the visual appearance of the results.